# Impacts of Thermal Processing, High Pressure, and CO_2_-Assisted High Pressure on Quality Characteristics and Shelf Life of Durian Fruit Puree

**DOI:** 10.3390/foods11172717

**Published:** 2022-09-05

**Authors:** Zhibin Bu, Wenshan Luo, Jiayin Wei, Jian Peng, Jijun Wu, Yujuan Xu, Yuanshan Yu, Lu Li

**Affiliations:** Sericultural & Argi-Food Research Institute, Guangdong Academy of Agricultural Sciences/Key Laboratory of Functional Foods, Ministry of Agriculture and Rural Affairs/Guangdong Key Laboratory of Agricultural Products Processing, No. 133 Yiheng Street, Dongguanzhuang Road, Tianhe District, Guangzhou 510610, China

**Keywords:** durian fruit puree, different treatments, quality properties, shelf life, storage

## Abstract

Durian fruit puree (DFP) is a nutrient-dense food, but it has a short shelf life. Presently, little research has been undertaken on extending the shelf life of DFP. Hence, it is necessary to develop treatment methods that can prolong the shelf life of DFP. In the present study, thermal processing (TP), high-pressure processing (HPP), and CO_2_-assisted HPP (CO_2_ + HPP) treatments are used for DFP, and their influences on quality properties of DFP during storage (35 days, 4 °C) are investigated. Compared to other treatments, the CO_2_ + HPP treatment had a lower pressure and a shorter time to achieve the same effect of inactivating the microorganisms of DFP. During storage, CO_2_ + HPP treated DFP showed higher retention rates of sugars, total soluble solids, color, bioactive components, and antioxidant capacity in comparison with other treated DFPs. Moreover, after 35 days of storage, the microbial count of (CO_2_ + HPP)-treated DFP (3.80 × 10^3^ CFU/g) was much lower than those of TP (4.77 × 10^5^ CFU/g) and HPP (8.53 × 10^3^ CFU/g)-treated DFPs. The results of this study reveal that CO_2_ + HPP treatment could not only better preserve the quality of DFP, but also effectively extend the shelf life of DFP, providing an effective method for the processing of DFP.

## 1. Introduction

Durian (*Durio zibethinus* Murr.) is a tropical fruit plant that has been cultivated in Southeast Asia for centuries. It is known as the ‘king of fruits’, owing to its unique taste and flavor. Durian fruit is rich in protein, dietary fiber and carbohydrates, resulting in abundant bioactive functions. Nowadays, the durian fruit is often eaten fresh or used as a flavor-enhancing ingredient [1]. Each durian fruit contains shin, seed, husk, and pulp, and the pulp accounts for 50–65% of the durian fruit [2]. Due to the short shelf life of durian pulp, it needs to be eaten or processed immediately [1]. Durian fruit puree (DFP) is a kind of durian pulp product that has been widely used in bakery products [3]. However, similar to durian pulp, the shelf life of DFP is very short, hindering its application.

In order to prolong the shelf life of food products, various processing methods have been used in the food industry [4]. Among them, thermal processing (TP) is the most widely applied [5]. In recent years, consumers have demanded fresh, high-quality and natural-flavored fruit products [6]. However, previous studies have reported that TP has a large negative influence on the bioactive compounds and flavor of fruit products [7]. Durian is a heat-sensitive fruit; hence, it can be inferred that TP is not suitable for the processing of durian products.

Non-thermal treatment methods, such as high-pressure processing (HPP) and CO_2_ treatment, have shown their potential for food processing [8]. HPP is a commercial technology that extends the shelf life of food by causing the inactivation of spoilage or pathogenic microorganisms in food through high pressure. Numerous studies have shown that HPP can be widely used to treat vegetable and fruit products [9,10]. Compared to TP, HPP treatment retains more physicochemical properties, active ingredients and flavors of fruit products [11]. However, HPP treatment has poor influence when inactivating microorganisms in low-acid food, which limits its application [12]. In recent years, CO_2_ treatment has presented the capacity to inactivate the microorganisms of fruit products [13]. The microbial inactivation mechanism caused by CO_2_ treatment is mainly to damage the cell membrane and reduce the pH of the cell [14]. Hence, CO_2_ treatment may be effective at inactivating microorganism in low-acid food. Moreover, CO_2_ can be used as a pretreatment to enhance the microorganism inactivation effect of HPP [15]. Considering that DFP is a neutral food (pH = 6.8–7.0), it can be speculated that CO_2_-assisted HPP treatment may be more suitable for the processing of DFP in comparison with HPP treatment. However, to our knowledge, there are no studies on CO_2_-assisted HPP-treated DFP. Therefore, in this study, TP, HPP, and CO_2_ + HPP treatments are used to treat DFP, and the influences of these three treatments on microbial count, physicochemical properties, bioactive ingredients, and the antioxidant capacity of DFP during storage are investigated.

## 2. Material and Methods

### 2.1. Preparation of DFP

Fresh durian fruits (*D. zibethinus* Murr. D24 cultivar) with the same ripeness (eating-ripe stage) were purchased from a local market in Guangzhou, China. The durian fruits were dehusked, cored, packed in vacuum bags, and stored at −20 °C. Frozen durian pulp was thawed in ice water for 2–3 h before processing. A household blender was applied to mix the thawed durian pulp and water (1:1.4, *w*/*w*). Then, 240 g DFP was aliquoted into a 250 mL polyethylene terephthalate (PET) bottle (height 13 cm, diameter 5.6 cm). The DFP was kept at 4 °C before further treatment.

### 2.2. Processing Conditions

TP treatment was carried out by a constant-temperature water bath. The temperature of the sample was detected by a digital temperature probe TP101 (Xintai Microelectronics Technology Co., Ltd., Shenzhen, China) during TP treatment. Timing for the thermal treatment started when the sample temperature reached 95 °C, and the treating time was set at 15, 30, 45, and 60 min. After the aforementioned treatment, the DFP sample was cooled to room temperature with an ice water bath.

CO_2_ treatment was conducted according to a previous study [15]. The sample was stored in an ice bath, then carbon dioxide (99.9% purity, Benlai biological technology Co., Ltd., Guangzhou, China) was added at 2.2 L/min from the bottom of the sample until the pH no longer changed. Finally, the PET bottles were capped as quickly as possible. The treated samples were kept at 4 °C before further treatment.

Next, 57 L SHPP-57DZM-600 HPP processing equipment (Sanshuihe Technology Co., Ltd., Taiyuan, China) was used to treat the sample. The samples were treated with different pressures (300, 400, 500, and 600 Mpa) at 25 °C for 10 min. During HPP treatment, the rise rate of pressure was 200 Mpa/min, and the release time was 5 s. The holding time did not include rise time or the release time of pressure.

The DFP without treatment was set as the control, and three samples were prepared for each treatment group. The microbial counts of the control and treated DFP samples were analyzed to screen the optimal treatment condition.

### 2.3. Storage Condition

The DFP treated with TP (95 °C, 60 min), HPP (600 Mpa), and (CO_2_ + HPP) (600 Mpa) were known as the TP-treated group, HPP-treated group, and CO_2_ + HPP-treated group, respectively. Eighteen samples were prepared for each treatment group. After treatment, all samples were stored at 4 °C in the dark for 35 days. Additionally, the microbial counts, pH, total soluble solids, color parameter, apparent viscosity, sucrose, glucose, fructose, ascorbic acid, total phenolic, total flavonoid, total carotenoid, and antioxidant capacity of samples were detected at 0, 7, 14, 21, 28, and 35 days. Three samples from each treatment group were collected for detection at each time point.

### 2.4. Microbial Counts

Plate count agar (PCA, tryptone 5 g/L, yeast extract 2.5 g/L, glucose 1 g/L, agar 15 g/L, pH 7.0 ± 0.2) and Rose Bengal agar (RBA, peptone 5 g/L, glucose 10 g/L, potassium dihydrogen phosphate 1 g/L, magnesium sulfate 0.5 g/L, agar 15 g/L, rose Bengal 0.033 g/L, chloramphenicol 0.1 g/L) were purchased from Guangdong Huankai Microbiology Technology Co., Ltd. (Guangzhou, China). The numbers of aerobic bacteria and yeast plus mold were counted by PCA and RBA methods, respectively. Then, a 15 g sample was added into 135 mL 0.9% sterile saline solution and mixed by a shaker (250 rpm, 25 °C) for 5 min. Then, each serial dilution was performed with 0.9% sterile saline solution and 1 mL of each diluted solution (10^−1^–10^−5^) was spread onto the PCA and RBA plates, respectively. The number of total aerobic bacteria was detected from the PCA plate after being cultured at 37 °C for 48 h, and the number of yeast plus mold was measured from the RBA plate after being incubated at 30 °C for 72 h. Microbial count was expressed as colony-forming units per gram (CFU/g).

### 2.5. Detection of pH, Total Soluble Solids, and Color

A Metrohm744 pH meter (Metrohm Co. Ltd., Hongkong, China) was used to detect the pH of DFP. The total soluble solids of the DFP samples were measured by a digital refractometer (RP-101, Atago, Tokyo, Japan) at 25 °C. The color of the DFP was detected by an UltraScan VIS colorimeter (HunterLab Co., Reston, VA, USA) in the Hunter scale. White color standard was used to calibrate the colorimeter, and the output results were *L* (degree of lightness) and *b* (degree of yellowness). The test was repeated six times, and results were averaged.

### 2.6. Rheological Measurement

The rheological analysis of DFP was performed at 25 °C by a stress-controlled rheometer (AR500EX, TA Instruments, New Castle, DE, USA). The plate–plate geometry was 40 mm, and the range of shear rate was 0.01–100 s^−1^. The power-law model was applied to fit the flow curve of DFP using the formula:$$\eta =K\times \gamma^{n-1}$$
where K is the consistency index and *n* is the flow behavior index (when *n* < 1, the solution is pseudoplastic or shear-thinning fluid; when *n* = 1, the solution is a Newtonian fluid; for *n* > 1, the solution is a dilatant or shear-thickened fluid).

### 2.7. Determination of Sugar Component and Ascorbic Acid

The concentrations of fructose, glucose, and sucrose were analyzed using LC-20AT high-performance liquid chromatography (HPLC) with an evaporative light-scattering detector. The separation of the analytes was obtained with a Shodex Asahipak NH2P-50 4E (4.6 mm × 250 mm, 5 μm) column at 40 °C. The injection volume was 10 μL, 70% acetonitrile was used as the mobile phase, and the elution flow rate was 1 mL/min. The content of ascorbic acid was measured using LC-20AT HPLC with a photodiode array detector. Ascorbic acid was separated on an Agilent Zorbax SB-C18 (4.6 mm × 250 mm, 5 μm) at an injection volume of 10 μL with a 1 mL/min elution flow rate. The mobile phase was 0.1 mol/L pH = 2.7 (NH_4_)_2_HPO_4_ solution. The temperature of the column was 30 °C and the detection wavelength was set at 254 nm.

### 2.8. Determination of Total Phenolic, Total Flavonoid and Total Carotenoid

The total phenolic was extracted according to our previous study [16]: 1 mL extract solution was added with 2 mL Folin–Ciocalteu reagent and 2 mL 10% Na_2_CO_3_ solution. The mixture was reacted in the dark for 60 min, and the absorbance was read at 760 nm. The content of total phenolic was presented as gallic acid equivalent (GAE) g/kg DFP (fresh weight).

Then, 6 mL total phenolic extract solution was mixed with 1 mL NaNO_2_ solution. After standing for 5 min, 1 mL 10% Al(NO_3_)_3_ was added into the mixture. After 6 min, the mixture was added with 4 mL 1 mol/L NaOH solution, and the mixture was reacted at 45 °C for 10 min. Then, the mixture was centrifuged at 1000× *g* for 10 min, and the absorbance of the supernatant was measured at 505 nm. Rutin was used as the standard to express the total flavonoid content of DFP (fresh weight).

Spectrophotometric method was applied to analyze the total carotenoid content of DFP [17]. Then, 15 g DFP sample was mixed with 25 mL acetone. After 24 h of extraction in the dark, the mixture was centrifuged at 3000× *g* for 15 min. Then, the supernatant was transferred to a 500 mL separation funnel with 40 mL petroleum ether. Ultrapure water was used to remove acetone, and this procedure was repeated 2–3 times until the acetone was totally removed. Finally, the water in the extract solution was removed by anhydrous sodium sulfate. The absorbance of the extract solution was read at 450 nm. The total carotenoid concentration of DFP (fresh weight) was computed according to following equation:$$\mathrm{Total}\;\mathrm{carotenoid}\;\mathrm{concentration}\;\left(\mu g/g\right)=\frac{A\times V\left(\mathrm{mL}\right)\times 10^{4}}{A_{1\;\mathrm{cm}}^{1\%}\times P\left(\mathrm{mL}\right)}$$
where A, V, $A_{1\;\mathrm{cm}}^{1\%}$, and P are the absorbance, extraction solution volume, and β-carotene extinction coefficient in the petroleum ether and sample volume, respectively.

### 2.9. Measurement of Antioxidant Capacity

DPPH and ABTS assays were used to analyze the antioxidant capacity of DFP. The DPPH and ABTS assays were performed according to our previous studies [11,18].

#### 2.9.1. 2,2-Diphenyl-2-Picryl-Hydrazyl (DPPH) Radical Scavenging Capacity

Next, 400 μL total phenolic extract solution was mixed with 3.5 mL 0.14 mol/L DPPH solution, and the mixture was reacted in the dark for 30 min. Then, the absorbance of reaction mixture was read at 515 nm. The DPPH radical scavenging capacity of DFP was expressed as μM Trolox equivalents (TE)/100 g DFP (fresh weight).

#### 2.9.2. 2,2′-Azino-Bis (3-Ethylbenzothiazoline-6-Sulfonic Acid) (ABTS) Radical Scavenging Capacity

The ABTS radical scavenging capacity of DFP was performed according to the manufacturer’s instructions in the kit (Nanjing Jiancheng, Bioengineering Institute Co., Ltd., Nanjing, China). The result was presented as μM Trolox equivalents (TE)/100 g DFP (fresh weight).

### 2.10. Statistical Analysis

All analysis experiments were carried out at least in triplicate to compute the mean values and standard deviation (SD), and the significance of differences at the 95% confidence level was analyzed by one-way analysis of variance (ANOVA) and Tukey’s HSD test (SPSS 17.0).

## 3. Results and Discussion

### 3.1. Influences of Different Treatments on Microbial Counts of DFP

TP treatment is a common method for lengthening the shelf life of food; hence, the impact of TP treatment on the microbial counts of DFP was investigated. The initial counts of total aerobic bacteria and yeast plus mold in DFP samples were 1.35 × 10^5^ CFU/g and 3.11 × 10^3^ CFU/g, respectively. As shown in Appendix A, all the TP treatments could completely inhibit the yeasts plus molds of DFP, while only 95 °C–60 min treatment resulted in complete inhibition of aerobic bacteria in DFP. With the increase in treatment time, the inhibitory effect of TP treatment on aerobic bacteria was gradually improved. Nevertheless, TP treatment with a long time was required to achieve the goal of the complete inhibition of aerobic bacteria and yeast plus mold.

Although HPP has been widely used in the food industry, its ability to inactivate microorganisms in low-acid food is limited [14]. Considering that DFP is a low-acid food, conventional HPP treatment may have a poor effect on the inactivation of microorganisms. Therefore, CO_2_-assisted HPP treatment was used to process DFP, and its influence on microbial counts of DFP are shown in Figure 1. When the pressure of HPP treatment reached 300 Mpa, the number of total aerobic bacteria only decreased 8% (Figure 1A), while the yeast and mold were totally inhibited at the same pressure (Figure 1B), indicating that yeast and mold were less pressure-tolerant than bacteria, which is in line with a previous study [12]. In addition, compared to the HPP-treated group, the DFP treated with CO_2_ + HPP exhibited a better inhibition effect on aerobic bacteria when the treatment pressure was higher than 300 Mpa. As the treatment pressure reached 500 Mpa, the aerobic bacteria in the CO_2_ + HPP group were completely inactivated, while the HPP group need to be treated at 600 Mpa to achieve the same effect. Tan et al. also found that the HPP (600 Mpa) treatment could effectively inhibit the microorganisms in durian pulp [19]. The more efficient inactivation of microorganisms in the CO_2_ + HPP group could be attributed to the accumulation of CO_2_ on the surface of the cell and the penetration into the cell under high pressure, leading to membrane damage and cell death [20].

In addition, the influences of TP, HPP and CO_2_ + HPP on microbial counts of DFP during refrigerated storage were studied. As shown in Figure 2, the counts of total aerobic bacteria in TP- (95 °C, 60 min), HPP- (600 Mpa), and (CO_2_ + HPP)-treated (600 Mpa) DFPs all showed an upward trend during 35 days of storage, while there was no yeast or mold detected. Wu et al. also found the same phenomenon in TP- and HPP-treated pineapple juice [11]. The increase in total aerobic bacteria in DFP might be due to the recovery of the microorganisms damaged by sterilization treatment [21]. On the 35th day of storage, TP-treated DFP obtained the highest total aerobic bacteria count (4.77 × 10^5^ CFU/g), followed by HPP-treated DFP (8.53 × 10^3^ CFU/g). The results revealed that CO_2_ + HPP treatment could better control the microbe of DFP during storage in comparison with other treatments. The same result was also confirmed by Li et al.: the combined treatment of CO_2_ with HPP is an effective method to prolong the shelf life of low- or medium-acid food [22].

### 3.2. Influences of Different Treatments on pH, Total Soluble Solids, and Sugars of DFP

To further evaluate the application potential of CO_2_ + HPP treatment on DFP, the influences of TP, HPP and CO_2_ + HPP treatments on pH, total soluble solids, and sugars of DFP during storage were investigated. Firstly, the changes in pH and total soluble solids in different treated DFPs during the storage period were studied. Compared to untreated DFPs, the pH of all treated groups was reduced, while their total soluble solid contents were all increased (Table 1). In the early period of storage, the pH and total soluble solids of treated groups remained stable, which is in accordance with previous studies [23]. However, the pH and total soluble solids of DFP exhibited significant changes with the emergence of microorganisms. Microbes can consume sugars and produce organic acid [24], resulting in a decrease in total soluble solids and pH in DFP.

In addition, after being treated with TP, HPP, or CO_2_ + HPP, the fructose and glucose concentrations of DFP were both decreased, while the sucrose concentration was increased (Table 1). The decrease in fructose and glucose could be attributed to the Maillard reaction, which was accelerated by TP and HPP treatment [25]. Among these three treated groups, the fructose and glucose contents of the CO_2_ + HPP group were higher than those of the other treated groups. Therefore, it could be deduced that a weaker Maillard reaction was obtained in the CO_2_ + HPP group in comparison with the other groups. After being treated with HPP, the cell integrity of DFP was lost, inducing the release of molecules from compartmentalized structures and non-covalent binding [26]. Hence, the sucrose concentrations of HPP- and (CO_2_ + HPP)-treated DFPs were both increased. During storage, the fructose, glucose, and sucrose concentrations of different treated DFPs exhibited varying degrees of decline. Additionally, the decrease in sugar content of DFP was mainly focused on the stage when microorganisms appeared.

### 3.3. Influence of Different Treatments on Color Parameters of DFP

Color has a large impact on the acceptability of food, and is a critical factor evaluated by consumers [27]. In order to comprehensively study the influence of treatments on DFP quality, the changes in lightness (*L*) and yellowness (*b*) of DFP during storage were studied. The *L* and *b* values of different DFPs were presented in Table 1. Compared to untreated DFP, the *L* and *b* values of treated DFP were both decreased. It is well known that TP treatment has a negative influence on the color of processed fruit products [9]; hence, the *L* and *b* values of the TP-treated DFP were the lowest among the treated groups. Though the HPP treatment also exhibited a negative effect on the *L* and *b* values of DFP, the impact was much lower than that of TP treatment. The result was in relation to a previous study that the *L* and *b* values of TP-treated blueberry puree were lower than that of HPP-treated blueberry puree [28]. CO_2_ treatment can inactivate poly-phenoloxidase and peroxidase in foods [29], which explains the highest *L* and *b* values obtained in (CO_2_ + HPP)-treated DFP. During 35 days of storage, the *L* and *b* values of all treated groups exhibited a downward trend, especially at the stage of microbial appearance, which was probably because the microorganisms produced some substances that had a negative effect on the lightness and yellowness of DFP [30].

### 3.4. Influence of Different Treatments on Apparent Viscosity of DFP

The rheological property of fruit puree is an important factor affecting its quality [31]; thus, the influence of different treatments on DFP’s apparent viscosity was studied. The apparent viscosity of all DFP samples decreased with increasing shear rate (Figure 3), which was in agreement with a pseudo-plastic flow behavior. Meanwhile, as shown in Appendix A, after viscosity fitted, *n* values of all DFP samples were below 1, indicating that the DFP was a pseudoplastic fluid. Similar results for rheological characteristics were found in another puree sample [31]. At low shear rates (0.01 to 1 s^−1^), significant differences were obtained among TP, HPP, CO_2_ + HPP, and untreated groups, while there were no remarkable differences at high shear rates. Among these DFPs, treated DFPs exhibited a lower apparent viscosity than untreated DFP, and the (CO_2_ + HPP)-treated DFP gained the lowest apparent viscosity (Figure 3A). The apparent viscosity of fruit puree is positively correlated with its polysaccharide content [32], and the polysaccharide content of DFP with CO_2_ + HPP treatment (14.13 g/kg) was lower than those of other groups (untreated 22.62 g/kg, TP 19.23 g/kg, HPP 16.11 g/kg), resulting in the lowest apparent viscosity being obtained in the (CO_2_ + HPP)-treated DFP. During the storage, the apparent viscosity of the CO_2_ + HPP group was always lower than those of the other treated groups. Additionally, the difference in apparent viscosity among these treated groups was further increased when microorganisms appeared. This phenomenon could be attributed to the increase in microorganisms in DFP with the ability to synthesize polysaccharides, resulting in an increase in the polysaccharide content of DFP [33].

### 3.5. Influence of Different Treatments on Bioactive Compounds of DFP

Phenolics are natural antioxidants existing widely in fruits [34]. Hence, the change in total phenolics of different treated DFPs during storage is shown in Figure 4A. The total phenolic content of DFPs under different treatments were all decreased, and the TP group exhibited the lowest retention rate in total phenolic content. This finding is in accordance with a study that found that HPP-treated carrot puree could better retain total phenolics than the TP group [35]. During storage, the total phenolic concentration of DFP increased with the advent of microorganisms, which is consistent with a previous study [16]. The increase in the phenolic concentration of DFP by microorganisms might be because they could convert some substances of DFP into phenolic compounds. At the end of storage, TP-treated DFP obtained the highest total phenolic concentration (168.13 mg GAE/L), much higher than that of the other groups.

Flavonoids are the most abundant polyphenols in the human diet, and exist widely in durian fruit [3]. Similar to total phenolic content, the total flavonoid contents of different treated groups were also lower than that of the untreated group (Figure 4B). Among these three treated groups, the CO_2_ + HPP group showed the best retention rate of total flavonoids (95%), followed by the HPP group (90%). TP treatment showed a significant negative influence on the total flavonoids of DFP, and the total flavonoid retention rate of TP-treated DFP was only 80%. Similar trends were found in total flavonoid concentrations of mulberry juice and multi-vegetable smoothies after TP and HPP processing [36,37]. During storage, the total flavonoid contents of different treated groups remained stable until the appearance of microorganisms. After storage for 35 days, the CO_2_ + HPP group was still found to be the best among these three groups, since it maintained 86% total flavonoids.

The ascorbic acid concentration of fruit products is often regarded as a crucial index for judging oxidative deterioration [38]. As shown in Figure 4C, all treatments showed a negative correlation with the ascorbic acid concentration of DFP. The ascorbic acid retention rates of TP, HPP, and CO_2_ + HPP groups were 69%, 81%, and 82%, respectively, indicating that HPP and CO_2_ + HPP treatments could better retain the ascorbic acid of DFP. A similar result was also reported by previous studies that the ascorbic acid content of HPP-treated products was much higher than that of TP-treated products [11,39]. In addition, decline tendencies were found in the ascorbic acid content of DFPs during storage. After storage, the ascorbic acid concentrations of TP, HPP, and CO_2_ + HPP groups decreased by 53%, 46%, and 39%, respectively. The ascorbic acid concentration of kiwifruit juice also decreased gradually with storage time [9].

Carotenoid is a major bioactive compound in durian fruit [3], and has been reported to have the ability of reducing the incidence of various cancers and cardiovascular disease [17]. As can be seen in Figure 4D, the total carotenoid content of DFP was increased after being treated with HPP or CO_2_ + HPP. Additionally, the highest total carotenoid content (27.71 μg/100 g) was found in (CO_2_ + HPP)-treated DFP. HPP treatment could lead to the irreversible protein denaturation of protein–carotenoid complexes, resulting in the release of carotenoids [40]. Hence, the total carotenoid concentration of HPP-treated DFP was increased. However, TP treatment presented a negative relationship with the total carotenoid content of DFP. Meanwhile, the total carotenoid contents of all treated DFPs remained stable during storage, which is similar to previous research [11]. After storage, (CO_2_ + HPP)-treated DFP still had the highest concentration of total carotenoids.

### 3.6. Influence of Different Treatments on Antioxidant Ability of DFP

The antioxidant capacity of fruit products is one of the important indexes to reflect their quality [41]. The influence of different processing treatments on the antioxidant capacity of DFP is shown in Figure 5. DPPH and ABTS methods were applied to evaluate the antioxidant capacity of DFP. Both the DPPH and ABTS assay results presented that the antioxidant capacities of TP-, HPP-, and (CO_2_ + HPP)-treated DFPs were all lower than that of untreated DFP. The lowest antioxidant capacity was detected in TP-treated DFP. Sulaiman et al. also found that both TP and HPP treatments could decrease the antioxidant capacity of fruit products, while HPP treatment maintained greater antioxidant activity than TP [42]. A direct correlation was achieved between bioactive compound content and antioxidant capacity, which suggests that (CO_2_ + HPP)-treated DFP had the highest antioxidant activity possibly due to it preserved high content of bioactive compounds. During storage, all the treated groups showed downward trends in antioxidant capacity. Compared with TP- and HPP-treated DFPs, (CO_2_ + HPP)-treated DFP showed a higher antioxidant ability during the storage period.

## 4. Conclusions

In summary, the effects of TP, HPP and CO_2_ + HPP treatments on quality properties and shelf life of DFP during storage were evaluated. The results of the single-factor experiment indicated that all the treatments could completely induce the microbial inactivation of DFP, while the CO_2_ + HPP treatment used lower treatment pressure (500 Mpa) and shorter treatment time (10 min) to obtain the same effect. Among these treatments, the CO_2_ + HPP treatment showed the best ability for preserving the color (*L* value 91%, *b* value 85%), fructose (97%), glucose (91%), total phenolic (95%), total flavonoid (95%), ascorbic acid (82%), and antioxidant capacity (DPPH 84%, ABTS 93%) of DFP. In addition, according to the results of the storage test, the microbial count of CO_2_ + HPP-treated DFP was always lower than those of the other treated DFPs. Compared to TP and HPP treatments, CO_2_ + HPP treatment could better maintain the physicochemical properties, active substances, and antioxidant capacity of DFP during storage. Therefore, CO_2_ + HPP treatment could serve as a potential nonthermal processing method for heat-sensitive and low-acid fruit products.

## Figures and Tables

**Figure 1 foods-11-02717-f001:**
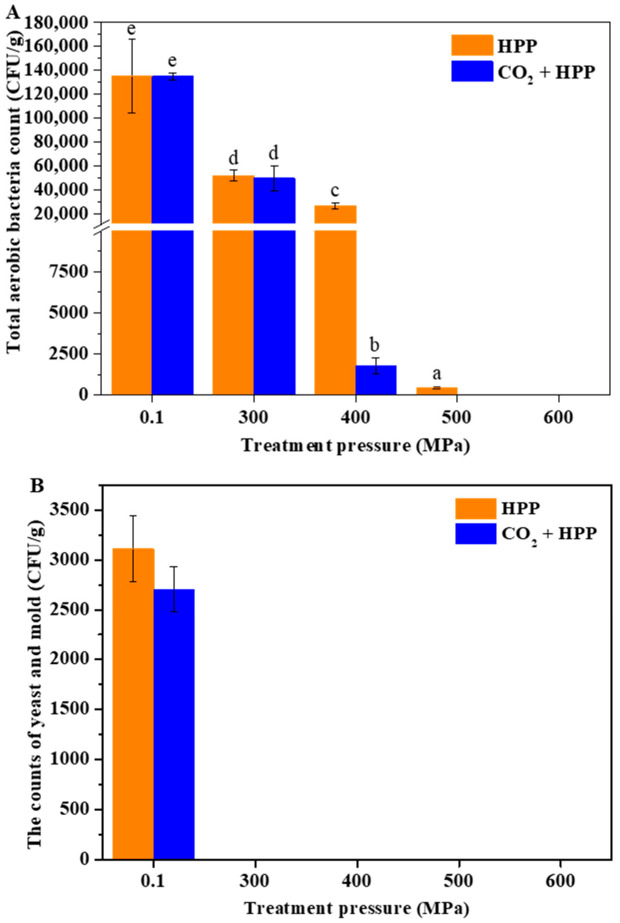
Effects of HPP and CO_2_ + HPP treatments with different pressures on total aerobic bacteria (**A**) and yeast plus mold (**B**) counts of DFP. Different letters (a, b, c, etc.) indicate significantly different means at *p* < 0.05 (analysis of variance (ANOVA)).

**Figure 2 foods-11-02717-f002:**
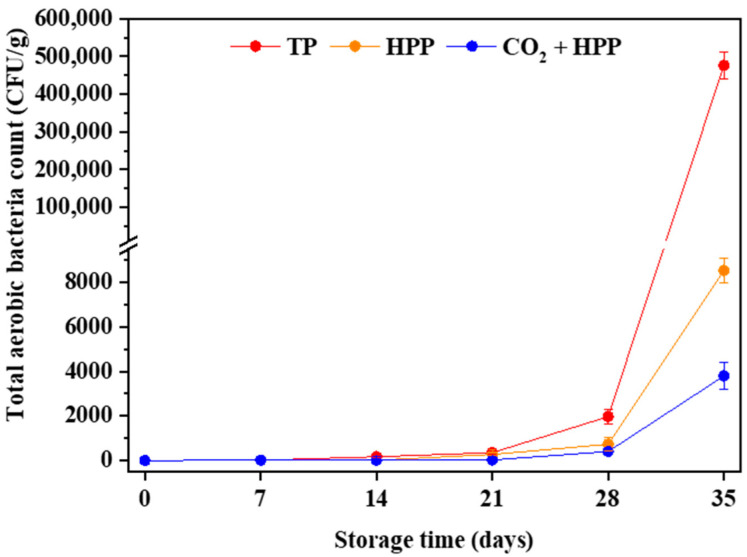
Effects of TP (95 °C, 60 min), HPP (600 Mpa) and CO_2_ + HPP (600 Mpa) treatments on total aerobic bacteria counts of DFPs during storage.

**Figure 3 foods-11-02717-f003:**
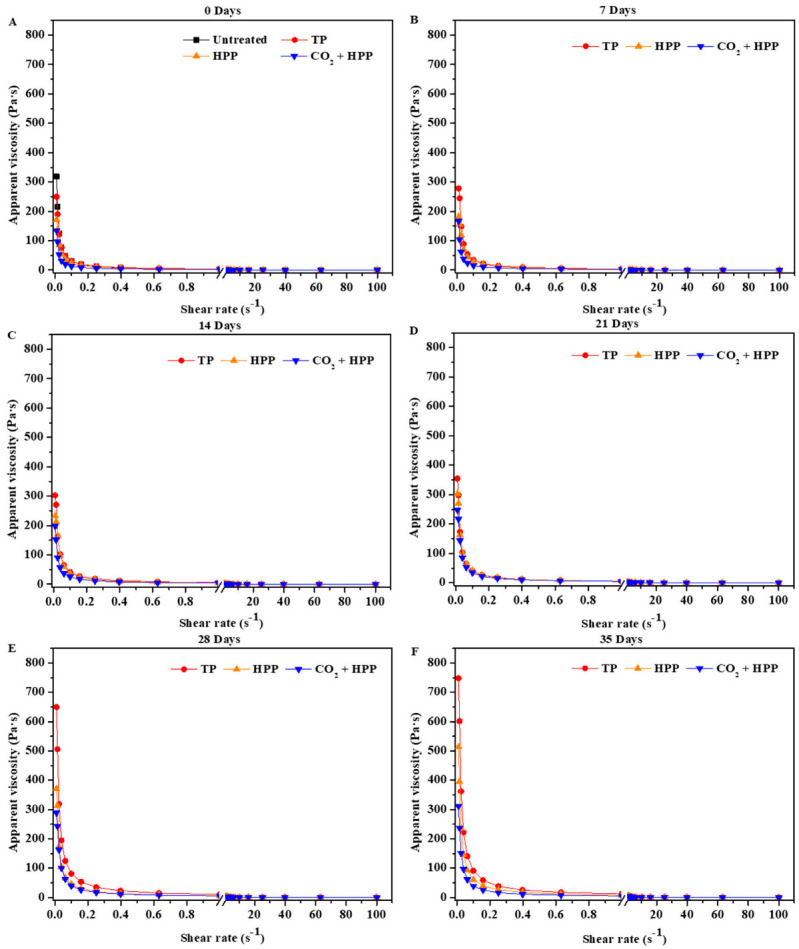
Changes in apparent viscosity of TP—(95 °C, 60 min), HPP—(600 Mpa), and (CO_2_ + HPP)—treated DFP (600 Mpa) during storage. (**A**–**F**) present 0, 7, 14, 21, 28, and 35 days, respectively.

**Figure 4 foods-11-02717-f004:**
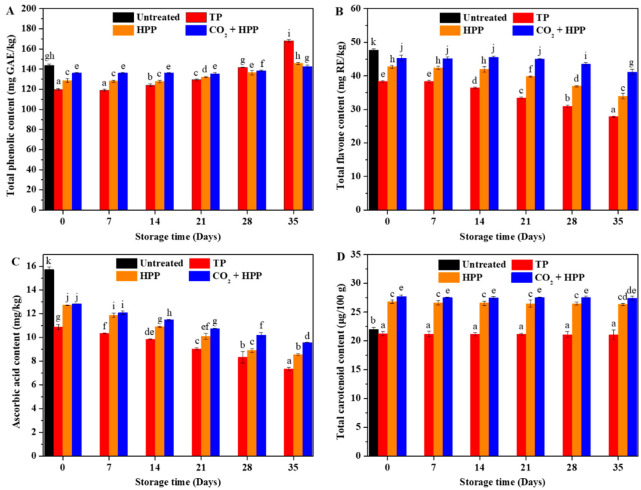
Analysis of total phenolic (**A**), total flavonoid (**B**), ascorbic acid (**C**), and total carotenoid (**D**) contents of TP—(95 °C, 60 min), HPP—(600 Mpa), and (CO_2_ + HPP)—treated DFP (600 Mpa) during storage. Different letters (a, b, c, etc.) indicate significantly different means at *p* < 0.05 (analysis of variance (ANOVA)).

**Figure 5 foods-11-02717-f005:**
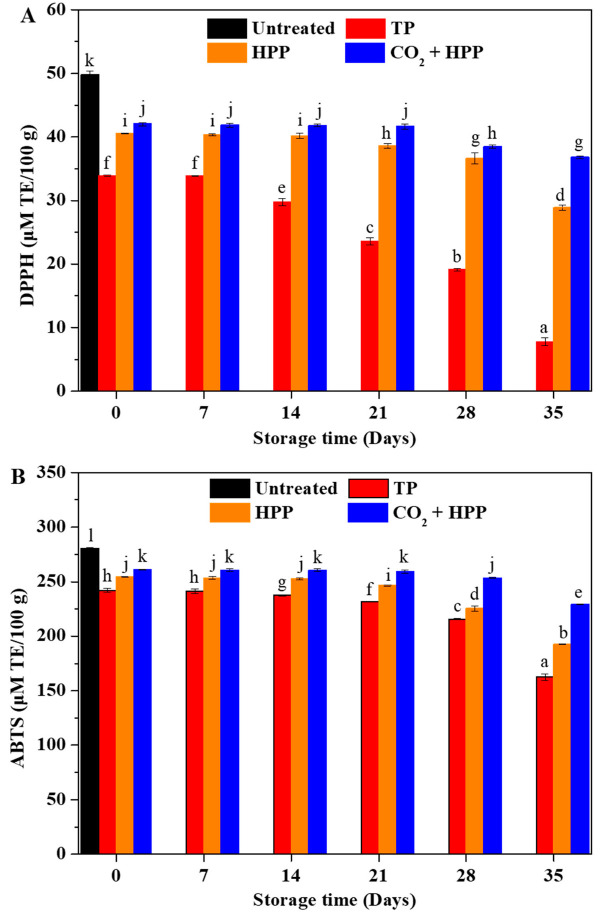
Analysis of DPPH antioxidant capacity (**A**) and ABTS antioxidant capacity (**B**) of TP- (95 °C, 60 min), HPP- (600 Mpa), and (CO_2_ + HPP)-treated DFP (600 Mpa) during storage. Different letters (a, b, c, etc.) indicate significantly different means at *p* < 0.05 (analysis of variance (ANOVA)).

**Table 1 foods-11-02717-t001:** Effects of TP (95 °C, 60 min), HPP (600 Mpa) and CO_2_ + HPP (600 Mpa) treatments on pH, total soluble solids, sugars, and color parameters of DFPs during storage.

Treatments	Storage Time (Days)	pH	Total Soluble Solid (°Brix)	Sugar Component (g/kg)			Color Parameter	
				Fructose	Glucose	Sucrose	*L*	*b*
Untreated	0	6.82 ± 0.02 i	9.01 ± 0.03 c	9.42 ± 0.02 l	8.47 ± 0.10 k	7.97 ± 0.18 a	69.98 ± 0.27 j	20.38 ± 0.27 h
TP	0	6.57 ± 0.02 g	9.86 ± 0.04 gh	5.65 ± 0.06 e	4.64 ± 0.06 e	39.19 ± 1.58 g	61.19 ± 0.14 g	15.41 ± 0.26 e
	7	6.56 ± 0.01 g	9.85 ± 0.04 g	5.53 ± 0.11 e	4.62 ± 0.01 e	38.49 ± 0.12 g	60.92 ± 0.36 g	15.36 ± 0.23 e
	14	6.36 ± 0.01 f	9.71 ± 0.04 f	5.30 ± 0.04 cd	4.41 ± 0.01 d	36.01 ± 0.11 f	59.79 ± 0.16 f	14.29 ± 0.18 d
	21	6.15 ± 0.04 e	9.48 ± 0.04 e	5.11 ± 0.03 c	4.20 ± 0.05 c	32.67 ± 0.89 e	58.84 ± 0.28 e	13.56 ± 0.20 c
	28	5.86 ± 0.05 d	9.05 ± 0.05 c	4.69 ± 0.07 b	3.88 ± 0.12 b	22.41 ± 3.78 c	56.82 ± 0.20 c	12.85 ± 0.14 b
	35	5.57 ± 0.02 a	8.10 ± 0.01 a	3.74 ± 0.10 a	3.12 ± 0.15 a	7.51 ± 0.10 a	52.81 ± 0.33 a	12.23 ± 0.10 a
HPP	0	6.64 ± 0.01 h	9.99 ± 0.08 j	7.19 ± 0.14 h	5.15 ± 0.03 g	34.80 ± 1.62 ef	62.85 ± 0.49 hi	16.40 ± 0.31 f
	7	6.63 ± 0.01 h	9.94 ± 0.04 hij	7.09 ± 0.09 h	5.09 ± 0.08 g	33.08 ± 0.90 e	62.43 ± 0.61 h	16.36 ± 0.33 f
	14	6.63 ± 0.01 h	9.93 ± 0.03 ghij	7.01 ± 0.20 h	5.06 ± 0.10 g	32.81 ± 0.94 e	62.23 ± 0.53 h	16.16 ± 0.25 f
	21	6.54 ± 0.01 g	9.72 ± 0.03 f	6.65 ± 0.10 g	4.85 ± 0.02 f	28.53 ± 0.20 d	60.61 ± 0.05 g	15.33 ± 0.17 e
	28	6.12 ± 0.01 e	9.30 ± 0.04 d	6.17 ± 0.08 f	4.67 ± 0.01 e	22.01 ± 0.91 c	58.20 ± 0.13 d	14.55 ± 0.25 d
	35	5.76 ± 0.03 c	8.56 ± 0.04 b	5.46 ± 0.03 de	4.35 ± 0.02 cd	15.12 ± 0.19 b	55.66 ± 0.14 b	13.18 ± 0.45 bc
CO_2_ + HPP	0	5.86 ± 0.01 d	9.95 ± 0.04 ij	9.18 ± 0.01 k	7.75 ± 0.16 j	35.02 ± 1.12 ef	63.45 ± 0.73 i	17.23 ± 0.13 g
	7	5.85 ± 0.01 d	9.90 ± 0.07 ghij	9.17 ± 0.15 k	7.73 ± 0.21 j	33.43 ± 1.22 e	62.90 ± 0.35 hi	17.13 ± 0.20 g
	14	5.84 ± 0.04 d	9.88 ± 0.08 ghi	9.10 ± 0.20 k	7.61 ± 0.20 j	33.22 ± 1.40 e	62.80 ± 0.20 h	17.06 ± 0.08 g
	21	5.83 ± 0.02 d	9.89 ± 0.05 ghi	9.07 ± 0.21 k	7.58 ± 0.17 j	32.73 ± 1.32 e	62.73 ± 0.24 h	17.04 ± 0.16 g
	28	5.71 ± 0.01 b	9.65 ± 0.03 f	8.77 ± 0.02 j	7.27 ± 0.01 i	28.50 ± 0.43 d	59.52 ± 0.50 f	16.36 ± 0.32 f
	35	5.59 ± 0.01 a	9.27 ± 0.05 d	8.14 ± 0.08 i	6.61 ± 0.01 h	21.27 ± 0.31 c	57.61 ± 0.24 d	15.53 ± 0.33 e

Data are expressed as means ± standard deviation (*n* = 3); different letters (a, b, c, etc.) indicate significantly different means at *p* < 0.05 (analysis of variance (ANOVA)).

## Data Availability

The data presented in this study are available on request from the corresponding author.

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
