# Peer review of "Impacts of Thermal Processing, High Pressure, and CO2-Assisted High Pressure on Quality Characteristics and Shelf Life of Durian Fruit Puree"

_foods, 2022, doi:10.3390/foods11172717_

Round 1
Reviewer 1 Report
This manuscript has investigated the influence of thermal processing (TP), high pressure processing (HPP), and high pressure processing in combination with CO2 (CO2 + HPP) on the physicochemical properties and bioactivity of durian puree during storage. The manuscript is well structured and the results are interesting.
Comments:
Line 77: Delete “respectively”
Line 82: Delete “respectively”
Line 86: Delete “respectively”
Line 151: Delete “respectively”
Line 155: tolerant
Line 231: The graphs do not show differences between treatments. Please fit the experimental data to the rheological models and report the parameters such as n and k.
Line 262: Why did the phenolic concentration of DFP increase with the advent of microorganisms? Add a reference.
Reviewer 2 Report
Comparison of thermal processing, high pressure and CO2-as-sisted high pressure on quality characteristics of durian fruit puree during storage is very interesting and well written.
Presented manuscript is on good scientific level and represent a very high scientific value manuscript.
The summary. Authors give a short presentation of manuscript.
Page 1, line 11 If you use abbreviation first time in manuscript text please give as well it full name: DFP (durian fruits puree)
Introduction section.
The Introduction section includes all necessary information about examined objects and problems. Formatted aim and main hypotheses are good presented at the end of Introductions' section. The problem described in manuscript is a new and represent problems of quality and different technique use for durian fruits preservation.
Materials and method section
Page 2, line 68: My question to authors: how many durian fruits were collected for experimental purposes. How many fruits were obtained per one combination?
Page 3, line 105: please listed HPLC equipment modules used for sugar and vitamin C determination.
Results
All results are good presented.
The discussion section presents a good comparison of the obtained results with other results available in the data basis.
Presented conclusions are corresponding with all information presented via Authors’ in manuscript text.
General opinion: After carefully manuscript reading, I think, that presented manuscript is a very valuable. In my opinion Manuscript should do minor correction. Manuscript is good to Food journal.
